# OpenReview forum: "The Sign Estimator: Preference Modeling for LLM Alignment under Heterogeneity"
_ICML.cc/2026/Conference — ICML 2026 regular_

### Official Review · Reviewer_KBK1 · 2026-03-01

**Soundness:** 3
**Presentation:** 2
**Significance:** 2
**Originality:** 3
**Overall Recommendation:** 3
**Confidence:** 3

**Summary:**

The paper studied the heterogeneity in preference modeling in the LLM settings. The proposed sign estimator yielded a reward model that is ordinally consistent with the population-average utility. The paper provided theoretical guarantee and numerical experiments to show the advantages of the proposed method.

**Compliance With Llm Reviewing Policy:**

Affirmed.

**Final Justification:**

The rebuttal addressed my main concerns.

**Key Questions For Authors:**

1. In Equation (2), $\mathcal{L}_{0-1}(\cdot)$ is not defined when it is used. In second line of Equation (2), what is $\tilde{\mu}(\cdot)$?
    Similar issue happens in the definition of $\hat{\mu}^{Sign}$ between Assumption 3.4 and Corollary 3.5.

2. What is the difference between $\beta$ and $\theta$?

3. In the experiments, is only $\theta$ finetuned? Or $\phi(\cdot)$ is also finetuned?

4. In section 5.2, do the labels in the preference datasets are simulated according to the the used model or assumptions for the proposed method?

**Limitations:**

The downstream policy training is not included.

**Strengths And Weaknesses:**

Strengths:

1. The proposed method solves an important problem by using a simple sign estimator.
2. Theoretical results are provided for the proposed method.
3. Numerical experiments are conducted to validate the proposed method.

Weaknesses:
1. The experimental results in Table 1 are not convincing. The performance of proposed method is slightly better than RLHF. No confidence intervals of the metrics are provided.
2. In Section 5.2, the paper simulates the labels. If the simulation mechanism follows the used model and assumptions for the proposed methods, while the baseline methods do not satisfy these conditions, the results are not convincing.
3. Equation (2) is one of the most important formulas in this paper. But this equation is confusing.

---

> ### Author Rebuttal · Authors · 2026-03-31
>
> We thank the reviewer for their feedback.
>
> Q1: Apologies for the confusion; Equation (2) is broken into two lines. $\tilde{u}(\cdot)$ is the parameter over which we optimize (i.e., a utility function).  The definition notation $\triangleq$ was missing.
> We replicate the formula here:
> \begin{equation}
> \begin{aligned}
> \hat{u}^{sign}
> &\in argmin_{\tilde u \in \mathcal U} \mathcal   L(\tilde u)
> \end{aligned}
> \end{equation}
> where we define the binary loss function for every $\tilde{u}$ as:
> \begin{equation}
> \begin{aligned}
> \mathcal L(\tilde u) \triangleq -\mathbb{E}_{X_1,X_2,Y}\Bigl[
> (2Y-1)
> \mathrm{sign}\bigl(
> \tilde u(X_1) - \tilde u(X_2)
> \bigr)
> \Bigr] .
> \end{aligned}
> \end{equation}
> Here, $ \mathcal L(\tilde u)$ is the binary classification loss function that measure the signed agreement between the chosen label $(2Y-1)$ and the ordinal ranking between utilities $\mathrm{sign}\bigl(\tilde u(X_1) - \tilde u(X_2) \bigr)$. Intuitively, the loss function is incremented by 1 if there is a “disagreement” and decrement by 1 if there is an “agreement”. Our goal to minimize disagreement.
>
> Q2: We use $\theta$ as notation for the parameter (utility vector) and \beta as the randomized experiment: \beta \sim {\cal B} is a utility vector drawn at random from the population. This is the difference between a random variable and possible outcome values. Thank you for raising this; we will clarify in future versions and be more consistent.
>
> Q3: In the experiments, we only train a linear head ($\theta$ in the utility function $\theta^\top \phi\langle X\rangle$). We do not fine tune the embeddings $\phi(\cdot)$; they are provided by the last hidden layer of gpt-oss 20B. This approach facilitates the comparison of utility functions/reward models to the ground truth. We can simply measure angles between preference vectors (i.e., angular loss). Otherwise comparing utility functions and evaluating the quality of recovery may not be straightforward.
>
> Q4: No, we do not use impose any of the assumptions required by our analysis. Our synthetic experiment faithfully simulates preferences from the panel of digital twin personas without any additional assumption. To reflect that, we estimate whether the symmetry assumption holds on that data set. We conduct a Kolmogorov-Smirnov test
> on random projections. The symmetry assumption is violated (bootstrap p<0.002) with a test statistic of 0.233 indicating significant levels of asymmetry. Despite that, the Sign Estimator achieves significantly higher performance.
>
> We kindly would like to comment on weaknesses:
>
> W1: We apologize for not providing confidence intervals for the original set of experiments. They are now reported in Table 1. All differences in performance between the Sign Estimator and RLHF are statistically significant  (non-overlapping 95% CIs), and in all cases but one for the EM algorithm. Notice that the EM algorithms have high variance and achieve the worst average performance.
> We conducted additional experiments using Open AI CoVal (N=18,384) dataset as additional robustness:
>
> Out-of-sample in-distribution: We observe consistent and statistically significant improvements in out-of-sample accuracy using train/validate/test splits on the Coval dataset.
>
> Generalization out-of-distribution. We fit the reward model on HH dataset then evaluate performance on Coval dataset. The performance gap with RLHF widens to 3-4%, suggesting our method generalizes better under a distribution shift.
>
> We confirm that our simulated benchmark does not impose the symmetry assumption (or any other assumption required for our analysis). Please see the response to Q4
>
> W2: Kindly see our response to Q1
>
> Table 1: Extended predictive numerical results. Values in parentheses denote half the width of 95% confidence intervals. Each experiment is repeated over 20 seeds (where we randomize data splits and each algorithm’s initialization).
>
> | Dataset | Sign (ours) | RLHF | EM* (K=2) | EM* (K=3) | EM* (K=5) |
> |---|---:|---:|---:|---:|---:|
> | SHP | **68.33 ± 0.25** | 66.50 ± 0.17 | 67.01 ± 1.77 | 60.83 ± 1.44 | 61.44 ± 1.12 |
> | HH-Anthropic | **68.58 ± 0.68** | 66.89 ± 0.58 | 65.01 ± 0.45 | 63.94 ± 0.80 | 64.19 ± 0.99 |
> | CoVal | **80.72 ± 0.24** | 78.09 ± 0.28 | 74.32 ± 0.60 | 73.99 ± 0.61 | 74.36 ± 0.58 |
> | CoVal (Trained on HH-Anthropic) | **73.07 ± 0.43** | 69.94 ± 0.62 | 65.29 ± 2.25 | 64.20 ± 2.76 | 64.55 ± 2.93 |
>
> *The EM algorithm is adapted from [1,2], see Appendix H for details.
>
> [1] Sriyash Poddar, Yanming Wan, Hamish Ivison, Abhishek Gupta, and Natasha Jaques. 2024. Personalizing reinforcement learning from human feedback with variational preference learning. Preprint, arXiv:2408.10075.
>
> [2] Souradip Chakraborty, Jiahao Qiu, Hui Yuan, Alec Koppel, Furong Huang, Dinesh Manocha, Amrit Singh Bedi, and Mengdi Wang. 2024. Maxmin-rlhf: Towards equitable alignment of large language models with diverse human preferences. arXiv preprint arXiv:2402.08925.

---

> > ### Author Rebuttal · Reviewer_KBK1 · 2026-04-02
> >
> > Thanks for the rebuttal. I have raised my score.

---

### Official Review · Reviewer_J88w · 2026-03-12

**Soundness:** 2
**Presentation:** 3
**Significance:** 3
**Originality:** 3
**Overall Recommendation:** 4
**Confidence:** 1

**Summary:**

This paper studies preference learning under heterogeneous human preferences. It argues that the standard RLHF reward-model objective based on cross-entropy can be misspecified in this setting and may produce a biased aggregate preference because it implicitly downweights users with more confident preferences. The paper proposes Sign Estimator, which replaces the standard loss with a sign-based objective. Under a symmetry assumption on preference heterogeneity, the estimator is shown to recover the population-average utility. Experiments on SHP, HH-Anthropic, and a semi-synthetic digital-twin benchmark show consistent improvements over RLHF and EM baselines.

**Compliance With Llm Reviewing Policy:**

Affirmed.

**Final Justification:**

The rebuttal has addressed my main concerns and I have raised my score.

**Key Questions For Authors:**

- How sensitive is the method to violations of the symmetry assumption, and is there a way to estimate whether this assumption approximately holds in practice?
- Can the authors report variance across seeds or statistical significance for the improvements on SHP and HH-Anthropic?
- Do the improvements in reward-model performance translate into better downstream RLHF policies?

**Limitations:**

yes

**Strengths And Weaknesses:**

Strengths
- Addresses an important and realistic issue in RLHF: heterogeneous human preferences, and clearly explains why the standard cross-entropy reward-model objective can be biased in this setting.
- The sign-based estimator that can act as a drop-in replacement for the standard reward-model loss, making it appealing for practical adoption
- Provides theoretical analysis and empirical evaluations across multiple datasets and they are consistent with the paper’s claims.

Weaknesses
- The theoretical guarantee relies on a symmetry assumption on the heterogeneity distribution, whose realism in real-world RLHF data is unclear.
- Empirical improvements on real datasets are modest, and the paper does not report statistical significance or variance across random seeds.

---

> ### Author Rebuttal · Authors · 2026-03-31
>
> We thank the reviewer for their constructive feedback.
>
> Q1: Our experiments do not impose the symmetry assumption (we use the real-world HH and SHP datasets), and our method still outperforms RLHF and the EM algorithm. To assess whether the symmetry assumption holds in practice, we conduct a Kolmogorov–Smirnov test on random projections. The assumption is violated in our digital twins experiments (bootstrap p < 0.002) with a test statistic of 0.233, indicating a substantial level of asymmetry.
>
> We conducted two additional experiments using the recently introduced OpenAI CoVal dataset (N = 18,384) as additional robustness:
>
> Out-of-sample in-distribution: We observe consistent and statistically significant improvements in out-of-sample accuracy using train/validate/test split on the CoVal data set.
>
> Generalization out-of-distribution. We train the reward model on HH dataset then evaluate performance of the fitted model on CoVal dataset. The performance gap with RLHF widens to 3-4%, suggesting our method generalizes better under a distribution shift.
>
> Q2: We apologize for not providing confidence intervals for the original set of experiments. They are now reported in Table 1 in our rebuttal to Reviewer KBK1. All differences in performance between the Sign Estimator and RLHF are statistically significant  (non-overlapping 95% CIs), and in all cases but one for the EM algorithm. Notice that the EM algorithms have high variance and achieve the worst average performance.
>
> Q3: While we do not test directly for downstream LLM performance, our work builds on existing literature where preference modeling is itself a meaningful performance criterion [1, 2, 3, 4].
>
> While 2% performance gap in preference accuracy seems modest, previous literature suggests that it can translate into a very important gap in real-world downstream performance. For example, [1] propose Nash Learning from Human feedback—a game-theoretic solution concept for alignment—and find that it achieves a 2% accuracy gain against RLHF—an improvement size comparable to our method. They evaluate downstream LLM performance and observe very significant LLM alignment gains; see Section G.1 in [1].
>
> We emphasize an important connection between our work and NLHF. We prove in Appendix G of the paper that under our assumptions, the Sign Estimator coincides with
>
> the NLHF max-min solution. The binary classification loss of the Sign Estimator aligns with the “winning probability” that defines a zero-sum game for NLHF.  In other words, the Sign Estimator provides an efficient implementation of NLHF with statistical guarantees for reward modeling. Notice that NLHF corresponds to the solution of a min-max problem, which is, in general, different from the Sign Estimator and may not generate a reward model; our approach is more computationally tractable and makes the reward model transparent.
>
> We briefly comment on the weaknesses:
>
> W1 The symmetry assumption is often satisfied by parametric econometric models. For example it is satisfied by the Gaussian mixed logits model, which is one of the most widely used choice models for heterogeneous preferences. We can relax this assumption to requiring that the median utility to be at the 1/2 probability level; another prevalent assumption in the econometrics literature[5,6].
>
> While our consistency recovery guarantees require this assumption, the Sign Estimator's practical performance does not hinge on it; see our response to Q1.
>
> W2: Confidence intervals are now reported in Table 1.  While accuracy gains of 1-2% seems modest, previous literature suggests that these gaps are sizeable. For example, a key prior study [1] proposed Nash Learning from Human feedback—a game-theoretic solution concept for alignment. They find that a 1-2% accuracy (in a simpler empirical set-up than ours) translates into significant downstream LLM performance gains.
>
> [1] Remi Munos, Michal Valko, et al. Nash learning from human feedback. In International Conference on Machine Learning, pages 36743–36768. PMLR, 2024.
>
> [2] Evan Frick, Tianle Li, Connor Chen, Wei-Lin Chiang, Anastasios N Angelopoulos, Jiantao Jiao, Banghua Zhu, Joseph E Gonzalez, and Ion Stoica. 2024. How to evaluate reward models for rlhf. arXiv preprint arXiv:2410.14872.
>
> [3] Enyu Zhou, Guodong Zheng, Binghai Wang, Zhiheng Xi, Shihan Dou, Rong Bao, Wei Shen, Limao Xiong, Jessica Fan, Yurong Mou, et al. Rmb: Comprehensively benchmarking reward models in llm alignment. arXiv preprint arXiv:2410.09893, 2024
>
> [4] Sriyash Poddar, Yanming Wan, Hamish Ivison, Abhishek Gupta, and Natasha Jaques. 2024. Personalizing reinforcement learning from human feedback with variational preference learning. Preprint, arXiv:2408.10075.
>
> [5] Ota, Y., & Otsu, T. (2026). Specification testing for binary choice model via maximum score. Economics Letters, 112929.
>
> [6]Blundell, R. W., & Powell, J. L. (2004). Endogeneity in semiparametric binary response models. The Review of Economic Studies, 71(3), 655-679.

---

> > ### Author Rebuttal · Reviewer_J88w · 2026-04-02
> >
> > Thank you for the clarifications. I have raised my score.

---

### Official Review · Reviewer_c2D1 · 2026-03-14

**Soundness:** 4
**Presentation:** 2
**Significance:** 3
**Originality:** 3
**Overall Recommendation:** 5
**Confidence:** 4

**Summary:**

The paper addresses the important yet understudied issue of reward modeling in RLHF, capturing the distributional preference. With theoretical analysis on the existing formulation of the classifier reward models, the paper proposes the sign estimator as a typical MLE-based reward modeling based on the Bradley-Terry model.

**Compliance With Llm Reviewing Policy:**

Affirmed.

**Final Justification:**

The concerns were mostly minor, and they were partially addressed. Further experiments and analysis over the practical preference benchmarks (e.g., RewardBench, RM-Bench) could strengthen the message of the paper. Thus, I remain with my positive score.

**Key Questions For Authors:**

- How would the sign estimator method impact the reward model benchmarks?

**Limitations:**

Not mentioned.

**Strengths And Weaknesses:**

**Strengths**
- The paper addresses the potential bias under heterogeneous preferences in the typical reward modeling objective, which is often overlooked in preference learning or the RLHF context.

- The theoretical analysis is compelling. In particular, the paper provides a clear account of how the standard RLHF estimator can bias aggregation toward uncertain users while discounting confident ones, which gives a concrete and intuitive explanation of the failure mode of the Bradley-Terry-style objective under heterogeneity.

- The experimental design well-aligns with the main claim of the paper, mitigating the utility direction error induced by the bias in the RLHF estimator.


**Weaknesses**

- While the main claim of the paper is in the misleading capturing of the heterogeneous preference with the typical RLHF reward modeling objective, the sign estimator still aims to accurately capture the preference hierarchies in the responses. Thus, additional wider evaluation on the existing reward model benchmarks, e.g., RewardBench or RM-Bench, would further support the empirical advantages of the proposed method.

- The related work discussion could be improved. There are prior works on alternative RLHF or preference-learning objectives that also question the adequacy of Bradley-Terry / MLE-based formulations from different perspectives [1, 2]. For example, KTO [2] approaches preference learning from the perspective of utility maximization rather than likelihood maximization. A broader discussion of how the present work relates to such alternatives would strengthen the positioning of the paper.

**Minor Suggestions**

- Hyperlink for the Appendix is broken in line 1219.

**References**

[1] Azar et al., 2023, "A General Theoretical Paradigm to Understand Learning from Human Preferences".

[2] Ethayarajh et al., 2024, "KTO: Model Alignment as Prospect Theoretic Optimization"

---

> ### Author Rebuttal · Authors · 2026-03-31
>
> We thank the reviewer for the thoughtful feedback.
>
> Our paper shows empirical advantages across various standard RLHF datasets (see Table 1). We agree that carefully evaluating our method on reward model benchmarks would be very valuable (the sign estimator can be used as a drop-in replacement for cross-entropy in reward model pipelines), and leave this for our future work.
>
> Still, we expanded our benchmark in the rebuttal and included a recently introduced annotation data set OpenAI CoVal (N=18,384); see Table 1 in our rebuttal to Reviewer KBK1. This dataset has up to 18 annotators per query. The Sign Estimator again achieves significantly higher accuracy out-of-sample. We also test generalization out of distribution: we train reward models on the HH data set and then evaluate their accuracy on CoVal. This evaluation procedure shows larger accuracy gains of 3-4%. This suggests that our method may be more robust to distribution shifts, often observed in real-world deployments.
>
> While accuracy gains of 1-2% seem modest, previous literature suggests that these gaps are sizeable. For example, a key prior study [3] proposed Nash Learning from Human feedback—a game-theoretic solution concept for alignment. They find that a 1-2% accuracy (in a simpler empirical set-up than ours) translates into significant downstream LLM performance gains. Notice that the Sign Estimator has interesting connections to NLHF. We prove in Appendix G of the paper that under our assumptions, the Sign Estimator coincides with the NLHF max-min solution. In other words, the Sign Estimator provides an efficient implementation of NLHF with statistical guarantees for reward modeling.
>
> We briefly comment on the weaknesses:
>
> W1: Thank you for suggesting a wider evaluation using Reward Model  Benchmarks. We focused on standard datasets to compare our Sign Estimator with RLHF and EM algorithms, cleanly isolating the effects of our new loss function and statistical estimator. This approach is used in the literature [4,5,6]. However, your suggestion is a valuable direction to explore for our future work and could provide additional empirical support.
>
> W2: Thank you for suggesting additional references to the literature on utility maximization in preference learning. We will include them in future versions of our paper. [1] provides a unifying theory for preference alignment, including RLHF and DPO. [2] integrates behavioral preference biases from prospect theory—offering a more general modeling of preferences. However, neither paper directly addresses the issue of heterogeneity.
>
> We view the unique contribution of our work as providing a practical estimator with provable recovery guarantees under heterogeneous preferences. In contrast, previous literature finds that utility distortions are often inevitable when aggregating heterogeneous annotators' preferences. While our theory requires some assumptions, it subsumes the (Gaussian) mixed logits model—one of the most frequently used models for heterogeneous preferences. We also expand our empirical evidence:
>
> (1) Our approach consistently dominates both RLHF and EM on the OpenAI CoVal dataset [7] (N = 18,384). We also test generalization performance out-of-distribution: We fit the reward model on HH and evaluate performance on CoVal. The performance gap widens to 3-4%, suggesting better generalization under a distribution shift.
>
> (2) A Kolmogorov-Smirnov test on random projections shows that the symmetry assumption is violated on the Anthropic HH digital twins experiments (bootstrap p<0.002) with a test statistic of 0.233, indicating a substantial level of asymmetry. Our method is efficient even when theoretical assumptions are violated.
>
> [1] Azar et al., 2023, "A General Theoretical Paradigm to Understand Learning from Human Preferences".
>
> [2] Ethayarajh et al., 2024, "KTO: Model Alignment as Prospect Theoretic Optimization"
>
> [3] Remi Munos, Michal Valko, Daniele Calandriello, Mohammad Gheshlaghi Azar, Mark Rowland, Zhaohan Daniel Guo, Yunhao Tang, Matthieu Geist, Thomas Mesnard, Come Fiegel, et al. Nash learning from human feedback. In International Conference on Machine Learning, pages 36743–36768. PMLR, 2024.
>
> [4] Evan Frick, Tianle Li, Connor Chen, Wei-Lin Chiang, Anastasios N Angelopoulos, Jiantao Jiao, Banghua Zhu, Joseph E Gonzalez, and Ion Stoica. 2024. How to evaluate reward models for rlhf. arXiv preprint arXiv:2410.14872.
>
> [5] Enyu Zhou, Guodong Zheng, Binghai Wang, Zhiheng Xi, Shihan Dou, Rong Bao, Wei Shen, Limao Xiong, Jessica Fan, Yurong Mou, et al. Rmb: Comprehensively benchmarking reward models in llm alignment. arXiv preprint arXiv:2410.09893, 2024
>
> [6] Sriyash Poddar, Yanming Wan, Hamish Ivison, Abhishek Gupta, and Natasha Jaques. 2024. Personalizing reinforcement learning from human feedback with variational preference learning. Preprint, arXiv:2408.10075.
>
> [7] OpenAI. 2025. CoVal: Public Input on Model Defaults (Version 2.0). [Data set].

---

> > ### Author Rebuttal · Reviewer_c2D1 · 2026-04-04
> >
> > Thank you for the further elaboration on the addressed points. I acknowledge the uniqueness of the work on studying the heterogeneous preferences, which is already shown through my positive score. With further study on the reward model benchmarks, the paper would be even more convincing with sufficient empirical support, which is not yet fully addressed through the rebuttal. Thereby, I remain with my positive score.

---

> > > ### Author Response · Authors · 2026-04-08
> > >
> > > We sincerely thank the reviewer for their constructive and helpful feedback and we are glad they enjoyed the uniqueness of our work. We have added more reward model benchmarks for empirical support in Table 1 in our response to Reviewer KBK1.

---

### Decision · Program_Chairs · 2026-04-30

**Decision:**

Accept (regular)

**Comment:**

The paper addresses the problem of preference modeling for LLM alignment under heterogeneous human annotators. The Authors propose the Sign Estimator, a simple drop-in replacement for the standard cross-entropy reward-modeling loss that instead minimizes a binary classification objective measuring ordinal agreement between model predictions and observed labels. Prior work shows recovering the population-average utility from RLHF is provably impossible in the worst case under preference heterogeneity, and the standard Bradley-Terry objective is known to be inconsistent, systematically down-weighting confident annotators in favor of uncertain ones. The proposed estimator addresses this by exploiting a symmetry assumption on the heterogeneity distribution, under which it is shown to be ordinally consistent with the population-average utility and to coincide with the Nash Learning from Human Feedback max-min solution. The Authors validate the approach on three benchmarks demonstrating statistically significant accuracy improvements together with a finite-sample convergence rate of $O(n^{−1/3})$ that avoids the curse of dimensionality.

There is broad consensus among the Reviewers that the problem is well-motivated and the theoretical contribution is solid. They also highlighted several issues such as the realism of the symmetry assumption (a Kolmogorov-Smirnov test on one of the benchmarks shows it is substantially violated), the modest empirical gains on real data without evaluation on standard reward model benchmarks or downstream policy performance, and notation and presentation issues in key formulas. These concerns were almost fully addressed (The one Reviewer who marked concerns as partially resolved maintained their positive score, noting that the remaining gap — lack of evaluation on established reward model benchmarks — is left for future work).

The paper is only slightly above the acceptance bar. If accepted, the Authors are kindly requested to properly revise and update the paper according to the raised concerns.